# The risk of *Plasmodium vivax* parasitaemia after *P. falciparum* malaria: An individual patient data meta-analysis from the WorldWide Antimalarial Resistance Network

Mohammad S. Hossain[1,2,3,4‡], Robert J. Commons[1,2,5‡], Nicholas M. Douglas[2,3], Kamala Thriemer[2], Bereket H. Alemayehu[6], Chanaki Amaratunga[7¤a¤b], Anupkumar R. Anvikar[8], Elizabeth A. Ashley[9,10], Puji B. S. Asih[11], Verena I. Carrara[9,12], Chanthap Lon[13,14], Umberto D'Alessandro[15], Timothy M. E. Davis[16], Arjen M. Dondorp[9,17], Michael D. Edstein[18], Rick M. Fairhurst[7¤c], Marcelo U. Ferreira[19], Jimee Hwang[20,21], Bart Janssens[22], Harin Karunajeewa[23,24], Jean R. Kiechel[25], Simone Ladeia-Andrade[26,27], Moses Laman[16,28], Mayfong Mayxay[9,10,29], Rose McGready[9,12], Brioni R. Moore[16,30], Ivo Mueller[31,32,33], Paul N. Newton[9,10,17], Nguyen T. Thuy-Nhien[34], Harald Noedl[35], Francois Nosten[9,12], Aung P. Phyo[12,36], Jeanne R. Poespoprodjo[37,38,39], David L. Saunders[40], Frank Smithuis[9,36,41], Michele D. Spring[13], Kasia Stepniewska[1,9], Seila Suon[42], Yupin Suputtamongkol[43], Din Syafruddin[11,44], Hien T. Tran[9,34], Neena Valecha[8], Michel Van Herp[22], Michele Van Vugt[12,17,45], Nicholas J. White[9,17], Philippe J. Guerin[1,9], Julie A. Simpson[1,3], Ric N. Price[1,2,9,17]*

1 WorldWide Antimalarial Resistance Network (WWARN), Oxford, United Kingdom, 2 Global Health Division, Menzies School of Health Research and Charles Darwin University, Darwin, Australia, 3 Centre for Epidemiology and Biostatistics, Melbourne School of Population and Global Health, The University of Melbourne, Melbourne, Victoria, Australia, 4 International Centre for Diarrheal Diseases and Research, Bangladesh (icddr,b), Dhaka, Bangladesh, 5 Internal Medical Services, Ballarat Health Services, Ballarat, Victoria, Australia, 6 ICAP at Mailman School of Public Health, Columbia University, New York, New York, United States of America, 7 Laboratory of Malaria and Vector Research, National Institute of Allergy and Infectious Diseases, National Institutes of Health, Rockville, Maryland, United States of America, 8 National Institute of Malaria Research, Dwarka, New Delhi, India, 9 Centre for Tropical Medicine and Global Health, Nuffield Department of Clinical Medicine, University of Oxford, Oxford, United Kingdom, 10 Lao-Oxford-Mahosot Hospital-Wellcome Trust Research Unit (LOMWRU), Microbiology Laboratory, Mahosot Hospital, Vientiane, Lao PDR, 11 Eijkman Institute for Molecular Biology, Jakarta, Indonesia, 12 Shoklo Malaria Research Unit, Mahidol-Oxford Tropical Medicine Research Unit, Faculty of Tropical Medicine, Mahidol University, Mae Sot, Thailand, 13 Department of Bacterial and Parasitic Diseases, Armed Forces Research Institute of Medical Sciences (AFRIMS), Bangkok, Thailand, 14 Armed Forces Research Institute of Medical Sciences, Phnom Penh, Cambodia, 15 Medical Research Council Unit The Gambia at LSTMH, Fajara, The Gambia, 16 Medical School, University of Western Australia, Fremantle Hospital, Fremantle, Australia, 17 Mahidol-Oxford Tropical Medicine Research Unit, Faculty of Tropical Medicine, Mahidol University, Bangkok, Thailand, 18 Australian Defence Force Malaria and Infectious Disease Institute, Enoggera, Brisbane, Australia, 19 Department of Parasitology, Institute of Biomedical Sciences, University of São Paulo, São Paulo, Brazil, 20 US President's Malaria Initiative, Malaria Branch, US Centers for Disease Control and Prevention, Atlanta, Georgia, United States of America, 21 Global Health Group, University of California San Francisco, San Francisco, California, United States of America, 22 Médecins Sans Frontieres, Brussels, Belgium, 23 Melbourne Medical School–Western Health, The University of Melbourne, Melbourne, Australia, 24 Western Health Chronic Disease Alliance, Sunshine Hospital, St Albans, Melbourne, Australia, 25 Drugs for Neglected Diseases initiative (DNDi), Geneva, Switzerland, 26 Laboratory of Parasitic Diseases, Oswaldo Cruz Institute/Oswaldo Cruz Foundation (Fiocruz), Rio de Janeiro, Brazil, 27 Amazonian Malaria Initiative/Amazon Network for the Surveillance of Antimalarial Drug Resistance, Ministry of Health of Brazil, Cruzeiro do Sul, Brazil, 28 Papua New Guinea Institute of Medical Research, Madang, Papua New Guinea, 29 Institute of Research and Education Development (IRED), University of Health Sciences, Ministry of Health, Vientiane, Lao PDR, 30 School of Pharmacy and Biomedical Sciences, Curtin University, Perth, Australia, 31 Division of Population Health and Immunity, The Walter & Eliza Hall Institute of Medical Research, Melbourne, Australia, 32 Department of Medical Biology, University of Melbourne, Melbourne, Australia, 33 Parasites and Insect Vectors Department, Institut Pasteur, Paris, France, 34 Oxford University

**Data Availability Statement:** The data are available for access via the WorldWide Antimalarial

Resistance Network (WWARN.org). Requests for access will be reviewed by a Data Access Committee to ensure that use of data protects the interests of the participants and researchers according to the terms of ethics approval and principles of equitable data sharing. Requests can be submitted by email to malariaDAC@iddo.org via the Data Access Form available at WWARN.org/accessing-data. The WWARN is registered with the Registry of Research Data Repositories (re3data.org).

**Funding:** MSH is supported by a Clinical Research and Development Fellowship scheme from TDR, the UNICEF/UNDP/World Bank/WHO Special Programme for Research and Training in Tropical Diseases. RJC is supported by a Postgraduate Australian National Health and Medical Research Council (NHMRC) Scholarship and a RACP NHMRC Kincaid-Smith Scholarship. RNP is a Wellcome Trust Senior Fellow in Clinical Science (200909). JAS is funded by an Australian NHMRC Senior Research Fellowship (1104975). KT is a CSL Centenary Fellow and received support by the Asia Pacific Malaria Elimination Network (APMEN) and OPRA clinical trial funding, supported by the Bill & Melinda Gates Foundation (INV-007122). PD is funded by Tropical Network Fund, Nuffield Department of Clinical Medicine, University of Oxford. WWARN is funded by Bill and Melinda Gates Foundation and Exxon Mobil Foundation grants. TMED is supported by an Australian Medical Research Future Fund Practitioner Fellowship. This work was supported by the Australian Centre for Research Excellence on Malaria Elimination (ACREME), funded by the NHMRC of Australia (1134989) and in part by the Intramural Research Program of the NIH, National Institute of Allergy and Infectious Diseases. The funders had no role in study design, data collection and analysis, decision to publish, or preparation of the manuscript.

**Competing interests:** I have read the journal's policy and the authors of this manuscript have the following competing interests: RMF is employed currently by AstraZeneca but has no financial stake in the results of the current study. EAA and NJW are Academic Editors on *PLOS Medicine*'s editorial board, and IM was previously an Academic Editor.

**Abbreviations:** AA, artesunate-amodiaquine; ACT, artemisinin-based combination therapy; AHR, adjusted hazard ratio; AL, artemether-lumefantrine; AM, artesunate-mefloquine; DP, dihydroartemisinin-piperaquine; G6PD, glucose-6-phosphate dehydrogenase; Hb, haemoglobin; HR, hazard ratio; IQR, interquartile range; MAP, Malaria Atlas Project; PNG, Papua New Guinea;

Clinical Research Unit (OUCRU), Ho Chi Minh City, Vietnam, **35** MARIB—Malaria Research Initiative Bandarban, Vienna, Austria, **36** Myanmar Oxford Clinical Research Unit, Yangon, Myanmar, **37** Mimika District Hospital, Timika, Indonesia, **38** Timika Malaria Research Programme, Papuan Health and Community Development Foundation, Timika, Indonesia, **39** Paediatric Research Office, Department of Child Health, Faculty of Medicine, Public Health and Nursing, Universitas Gadjah Mada/Dr. Sardjito Hospital, Yogyakarta, Indonesia, **40** Division of Medicine, United States Army Research Institute of Infectious Diseases, Ft. Detrick, Maryland, United States of America, **41** Medical Action Myanmar, Yangon, Myanmar, **42** National Center for Parasitology, Entomology, and Malaria Control, Phnom Penh, Cambodia, **43** Department of Medicine, Faculty of Medicine Siriraj Hospital, Mahidol University, Bangkok, Thailand, **44** Department of Parasitology, Faculty of Medicine, Hasanuddin University, Makassar, Indonesia, **45** Academic Medical Centre, Department of Internal Medicine, Slotervaart Hospital, Amsterdam, The Netherlands

¤a Current address: Mahidol-Oxford Tropical Medicine Research Unit, Faculty of Tropical Medicine, Mahidol University, Bangkok, Thailand
¤b Current address: Centre for Tropical Medicine and Global Health, Nuffield Department of Clinical Medicine, University of Oxford, Oxford, United Kingdom
¤c Current address: AstraZeneca, Gaithersburg, Maryland, United States of America
‡ These authors share joint first authorship on this work.
* ric.price@menzies.edu.au

# Abstract

## Background

There is a high risk of *Plasmodium vivax* parasitaemia following treatment of falciparum malaria. Our study aimed to quantify this risk and the associated determinants using an individual patient data meta-analysis in order to identify populations in which a policy of universal radical cure, combining artemisinin-based combination therapy (ACT) with a hypnozoitocidal antimalarial drug, would be beneficial.

## Methods and findings

A systematic review of Medline, Embase, Web of Science, and the Cochrane Database of Systematic Reviews identified efficacy studies of uncomplicated falciparum malaria treated with ACT that were undertaken in regions coendemic for *P. vivax* between 1 January 1960 and 5 January 2018. Data from eligible studies were pooled using standardised methodology. The risk of *P. vivax* parasitaemia at days 42 and 63 and associated risk factors were investigated by multivariable Cox regression analyses. Study quality was assessed using a tool developed by the Joanna Briggs Institute. The study was registered in the International Prospective Register of Systematic Reviews (PROSPERO: CRD42018097400). In total, 42 studies enrolling 15,341 patients were included in the analysis, including 30 randomised controlled trials and 12 cohort studies. Overall, 14,146 (92.2%) patients had *P. falciparum* monoinfection and 1,195 (7.8%) mixed infection with *P. falciparum* and *P. vivax*. The median age was 17.0 years (interquartile range [IQR] = 9.0–29.0 years; range = 0–80 years), with 1,584 (10.3%) patients younger than 5 years. 2,711 (17.7%) patients were treated with artemether-lumefantrine (AL, 13 studies), 651 (4.2%) with artesunate-amodiaquine (AA, 6 studies), 7,340 (47.8%) with artesunate-mefloquine (AM, 25 studies), and 4,639 (30.2%) with dihydroartemisinin-piperaquine (DP, 16 studies). 14,537 patients (94.8%) were enrolled from the Asia-Pacific region, 684 (4.5%) from the Americas, and 120 (0.8%) from Africa. At day 42, the cumulative risk of vivax parasitaemia following treatment of *P. falciparum* was

PROSPERO, International Prospective Register of Systematic Reviews; WHO, World Health Organization; WWARN, WorldWide Antimalarial Resistance Network.

31.1% (95% CI 28.9–33.4) after AL, 14.1% (95% CI 10.8–18.3) after AA, 7.4% (95% CI 6.7–8.1) after AM, and 4.5% (95% CI 3.9–5.3) after DP. By day 63, the risks had risen to 39.9% (95% CI 36.6–43.3), 42.4% (95% CI 34.7–51.2), 22.8% (95% CI 21.2–24.4), and 12.8% (95% CI 11.4–14.5), respectively. In multivariable analyses, the highest rate of *P. vivax* parasitaemia over 42 days of follow-up was in patients residing in areas of short relapse periodicity (adjusted hazard ratio [AHR] = 6.2, 95% CI 2.0–19.5; p = 0.002); patients treated with AL (AHR = 6.2, 95% CI 4.6–8.5; p < 0.001), AA (AHR = 2.3, 95% CI 1.4–3.7; p = 0.001), or AM (AHR = 1.4, 95% CI 1.0–1.9; p = 0.028) compared with DP; and patients who did not clear their initial parasitaemia within 2 days (AHR = 1.8, 95% CI 1.4–2.3; p < 0.001). The analysis was limited by heterogeneity between study populations and lack of data from very low transmission settings. Study quality was high.

## Conclusions

In this meta-analysis, we found a high risk of *P. vivax* parasitaemia after treatment of *P. falciparum* malaria that varied significantly between studies. These *P. vivax* infections are likely attributable to relapses that could be prevented with radical cure including a hypnozoitocidal agent; however, the benefits of such a novel strategy will vary considerably between geographical areas.

## Author summary

### Why was this study done?

- *Plasmodium vivax* is the most geographically widespread human malaria species; outside of sub-Saharan Africa, it almost invariably coexists with *P. falciparum*.

- A recent systematic review of 153 studies across 21 countries revealed that within 63 days of being treated for *P. falciparum* monoinfection, more than 15% of patients treated with an artemisinin-based combination therapy (ACT) had an episode of *P. vivax* malaria, far greater than expected for patients in similar locations who do not have acute malaria.

- We hypothesised that patients presenting with *P. falciparum* in coendemic locations are at high risk of carrying *P. vivax* dormant liver stages (hypnozoites) and would therefore potentially benefit from presumptive radical cure.

### What did the researchers do and find?

- We undertook an individual patient data meta-analysis to define the risk of vivax parasitaemia (blood stream infection) after treatment of *P. falciparum* malaria in different coendemic environments and to explore the factors underlying these risks.

- In total, 42 studies enrolling 15,341 patients were included in the analysis. By day 63, the risk of *P. vivax* ranged from 12.8% following dihydroartemisinin-piperaquine (DP) to 42.4% following artesunate-amodiaquine (AA).

- The highest rate of *P. vivax* malaria was in patients residing in areas with high risk of relapses and those who were slow to clear their initial parasitaemia.

## What do these findings mean?

- There is a high risk of vivax malaria after treatment of falciparum infection after all ACTs, although the risk varies substantially between locations.

- The correlation between the risk of *P. vivax* and the initial clearance of *P. falciparum* raises the possibility that the host response to acute malaria may be triggering the reactivation of *P. vivax* dormant liver stages.

- Universal radical cure with treatment to kill the liver stages may be warranted for reducing *P. vivax*, but the benefits will vary considerably between geographical locations, and further prospective clinical efficacy studies will be needed to determine the risks and benefits of such a strategy.

## Introduction

Malaria continues to exert a huge global health burden, with the latest estimates suggesting there are more than 228 million cases per year, associated with 405,000 deaths [1]. In 2015, the World Health Assembly set a target to reduce malaria prevalence by 90% by 2030 [2]. Although the burden of malaria has fallen in many regions, global estimates of malaria cases have plateaued and even risen in some areas over the past 5 years [1]. There has also been a relative rise in the proportion of malaria due to *Plasmodium vivax* outside of sub-Saharan Africa [3]. *P. vivax* is more difficult to eliminate than *P. falciparum* because it can form dormant liver stages (hypnozoites) that can reactivate weeks to months after the initial infection. Treatment of both the blood and liver stages is referred to as radical cure. If the proposed ambitious malaria elimination targets are to be achieved, innovative strategies are needed to provide safe and effective radical cure to the high proportion of individuals harbouring occult as well as circulating *P. vivax* parasites [4].

Where *P. vivax* and *P. falciparum* are coendemic (Asia, the Horn of Africa, and the Americas), reports have documented a high risk of *P. vivax* parasitaemia following treatment of *P. falciparum* infection [5, 6]. A recent meta-analysis of *P. falciparum* clinical trials revealed that within 63 days, 24% of patients presenting with *P. falciparum* had a recurrence with *P. vivax*, and almost 70% of all parasitological recurrences were due to *P. vivax* [7]. The high risk of *P. vivax* has been hypothesised to be due to reactivation of the dormant liver stages of parasites, although the mechanisms underlying this are not well understood. In areas where the risk of *P. vivax* parasitaemia is high, broadening the indication for hypnozoitocidal treatment to include patients presenting with uncomplicated malaria due to either *P. vivax* or *P. falciparum* (universal radical cure) has potential to reduce the subsequent risk of vivax parasitaemia and its ongoing transmission [8].

Radical cure with 8-aminoquinoline compounds can reduce the risk of *P. vivax* relapse significantly; however, both primaquine and tafenoquine can cause haemolysis in individuals with glucose-6-phosphate dehydrogenase (G6PD) deficiency [9]. Thus, the risk versus benefit

of a policy of universal radical cure will depend upon quantifying the inherent risk of *P. vivax* recurrence and associated cofactors in different endemic settings, an approach that is more amenable to an individual patient data meta-analysis than our previous study-level systematic review [7].

The aim of this individual patient data meta-analysis was to define the risk of vivax parasitaemia after falciparum infection in different coendemic environments and to explore the factors underlying these risks in order to identify populations in which a policy of universal radical cure, with blood schizontocidal treatment plus either primaquine or tafenoquine, would be most beneficial.

## Methods

### Search strategy and selection criteria

A previous systematic review was used to identify suitable studies [7]. In brief, Medline, Embase, Web of Science, and the Cochrane Database of Systematic Reviews were searched for prospective studies published between 1 January 1960 and 5 January 2018 in any language that included supervised treatment of patients with uncomplicated *P. falciparum* infection (including monoinfection and mixed *P. falciparum* and *P. vivax* infection) located in areas coendemic for *P. falciparum* and *P. vivax* (S1 Box). Countries were considered coendemic if indigenous *P. falciparum* and *P. vivax* cases were reported or suspected in 2016 [10].

In the original systematic review, studies were included if they explicitly reported the presence or absence of recurrent parasitaemia. Studies were excluded if patients were followed less than 28 days, if patients were only followed passively, or if the full text manuscript was unavailable. Because primaquine and tafenoquine are contraindicated in pregnant women and not a clinical priority in patients presenting with severe malaria, studies only enrolling these patients were also excluded from the analysis. Identification of studies and extraction of data were undertaken by 2 independent authors, with discrepancies resolved by discussion with a third author.

To improve the generalisability of the results, the inclusion criteria for this individual patient data meta-analysis were restricted further to include only studies in which patients were treated with 1 of 4 widely used ACTs: artemether-lumefantrine (AL), artesunate-amodiaquine (AA), artesunate-mefloquine (AM), or dihydroartemisinin-piperaquine (DP). Furthermore, because it was not always apparent whether recurrent *P. vivax* was systematically documented during follow-up, studies were only included if at least 1 episode of *P. vivax* parasitaemia during follow-up was reported in the manuscript. Additional study-level exclusions were if studies only included travellers or soldiers or only patients with hyperparasitaemic infection, because these either reflected malaria acquired from diverse locations or a biased population of patients attending a clinic in a malaria-endemic setting. Investigators of eligible studies were invited to participate in this analysis and share their study's individual patient data with the WorldWide Antimalarial Resistance Network (WWARN) repository. Shared data were anonymised and standardised using the WWARN Data Management and Statistical Analysis Plans [11].

In addition to the systematic review, the pre-existing WWARN repository was searched for studies of uncomplicated *P. falciparum* monoinfection or mixed infection in which at least 1 patient was recorded as having *P. vivax* parasitaemia during study follow-up. This was undertaken to identify unpublished studies and published studies that did not report *P. vivax* parasitaemia during follow-up in the manuscript. Patient data from eligible studies were collated together with those from the systematic review for analysis.

## Data collection and definitions

Demographic and clinical data including age, sex, weight, baseline species and parasite density, presence and species of parasites at follow-up, baseline gametocytaemia, baseline body temperature, and haemoglobin (Hb) or haematocrit at baseline and follow-up were included in the data set. Patient-level data were excluded if baseline parasitaemia was not available or did not include *P. falciparum* species, patients were treated with adjunctive antimalarials, or there was an enrolment violation according to supplied study data.

Haematocrit was converted to Hb using the formula Hb = (Haematocrit − 5.62)/2.6 [12]. The maximum value was used when multiple Hb measurements were recorded on a single day. The nadir Hb following acute *P. falciparum* malaria occurs between day 3 and 7, with most studies measuring the first follow-up Hb 7 days after starting antimalarial treatment [13]. The early decline and recovery of Hb was therefore assessed from the Hb measured between days 6 and 8 inclusive. Anaemia was defined as a Hb concentration less than 10 g/dL. A high parasite count was defined as a parasite count greater than 100,000 parasites per μL. Based on geographical location, studies were categorised into long or short *P. vivax* relapse periodicity using data from the Malaria Atlas Project (MAP) [14]. A median time to relapse of ≤47 days was categorised as short relapse periodicity. Subnational malaria parasite incidence was estimated from MAP models [15].

Data regarding the supervision of drug administration were extracted from publications or study protocols. Drug administration was categorised as 'supervised' if the entire treatment regimen was supervised, 'partially supervised' if only some treatment doses were supervised, and 'not specified if data were not available. The year of enrolment was defined in the first instance from individual patient data or, if unavailable, from the median year during which patients were enrolled into the study or as 2 years prior to the year of publication. The study was registered in the International Prospective Register of Systematic Reviews (PROSPERO: CRD42018097400), and the protocol is available at www.wwarn.org/working-together/study-groups/vivax-after-falciparum-study-group (S1 PRISMA Checklist).

## Outcomes

The primary outcome was the risk of *P. vivax* parasitaemia (*P. vivax* monoinfection or mixed infection) between days 7 and 42. Secondary outcomes were the risk of *P. vivax* parasitaemia between days 7 and 28 and days 7 and 63.

## Statistical analysis

Statistical analyses were undertaken using Stata software, version 15.1 (Stata Corporation, College Station, TX, USA) and R version 3.5.1 (R Foundation for Statistical Computing, Vienna, Austria), according to an a priori statistical analysis plan [16]. The cumulative risk of peripheral parasitaemia was calculated using Kaplan–Meier survival analyses. Patients were right-censored at their first recurrence, day of last follow-up, the day prior to a blood smear result gap >18 days, or the day of the planned outcome assessment [11]. Following peer review, prediction intervals were calculated for the cumulative risk of *P. vivax* parasitaemia [17].

To estimate the association between ACTs with the rate of *P. vivax* parasitaemia, Cox's proportional hazards regression was used with shared frailty for study sites and adjustment for age, sex, baseline parasitaemia, regional relapse periodicity, *P. falciparum* gametocytes, mixed infection at baseline, and Hb. The proportional hazards assumption was assessed visually using plots of observed versus expected values. Body weight and geographical location were not included because of collinearity with age and relapse periodicity, respectively. To investigate the association between the change in Hb and the rate of *P. vivax* parasitaemia, the

absolute fall in Hb between day 0 and day 7 was included in the multivariable Cox regression model.

The association between day of first parasite clearance (defined as parasite count below the microscopic limit of detection) and risk of *P. vivax* parasitaemia between days 7 and 63 was assessed by Cox's proportional hazards regression with shared frailty for study sites and adjustment for age, sex, baseline parasitaemia, regional relapse periodicity, *P. falciparum* gametocytes, mixed infection at baseline, Hb at baseline, and treatment. Patients treated with single low-dose primaquine were excluded. A subgroup analysis restricted data to patients treated with AL with a minimum of 42 days of follow-up to reduce the impact of slowly eliminated partner drugs with prolonged post-treatment prophylactic effect on outcome assessment. This subgroup was used to explore the association between the background incidence of *P. falciparum* and *P. vivax* and the risk of *P. vivax* parasitaemia. Following peer review, additional subgroup analyses were included for the other ACTs.

Bias related to individual studies was assessed using a tool developed by the Joanna Briggs Institute (S1 Text) [18] and in a sensitivity analysis, with removal of 1 study at a time and calculation of the coefficient of variation. Included and targeted studies that were not included were compared using baseline characteristics.

## Ethics

All data included in this analysis were obtained in accordance with ethical approvals from the country of origin. The data are fully anonymised and cannot be traced back to identifiable individuals. This systematic review did not require separate ethical approval according to the guidelines of the Oxford Central University Research Ethics Committee.

## Results

There were 153 clinical trials enrolling *P. falciparum* patients published between 1 January 1960 and 5 January 2018 and identified as part of the previous systematic review [7]. Of these, 51 (33.3%) studies enrolling 15,903 patients between 1991 and 2016 included treatment with the 4 major ACTs and were eligible for inclusion in this analysis. Individual patient data were available for 9,410 (59.2%) patients from 26 studies [19–44] (S1 Table and S2 Table). An additional 32 published and 2 unpublished studies with individual patient data in the WWARN repository were screened for inclusion, of which 16 studies were identified as also being eligible for the analysis [45–58]. In total, 42 studies enrolling 15,341 eligible patients were included in the analysis (Fig 1, S2 Table and S3 Table).

Of the studies included, 2,711 (17.7%) patients were treated with AL (13 studies), 651 (4.2%) with AA (6 studies), 7,340 (47.8%) with AM (25 studies), and 4,639 (30.2%) with DP (16 studies) (Fig 1 and S2 Table). In 2 studies, 448 patients were coadministered a single dose of primaquine. Patients were followed for 28 days in 3 studies (n = 670 patients), 42 days in 25 studies (n = 6,510 patients), 56 days in 1 study (n = 2,072 patients), 63 days in 11 studies (n = 5,653 patients), and a mixture of 28 and 63 days in 2 studies (436 patients).

The patients' baseline characteristics are presented in Table 1. Overall, 14,146 (92.2%) patients had *P. falciparum* monoinfection and 1,195 (7.8%) mixed infection with *P. falciparum* and *P. vivax* confirmed by blood film microscopy. The median age was 17.0 years (interquartile range [IQR] = 9.0–29.0 years; range = 0–80 years), with 1,584 (10.3%) patients younger than 5 years. Most patients (14,258; 92.9%) were recruited in areas with short relapse periodicity, and 82.5% (12,652) were enrolled in the Greater Mekong Subregion (S1 Fig). All 684 (4.5%) patients enrolled in the Americas were treated with AM or DP, whereas in Africa, all patients (120; 0.8%) were treated with AL. The 14,537 (94.8%) patients enrolled in Asia were

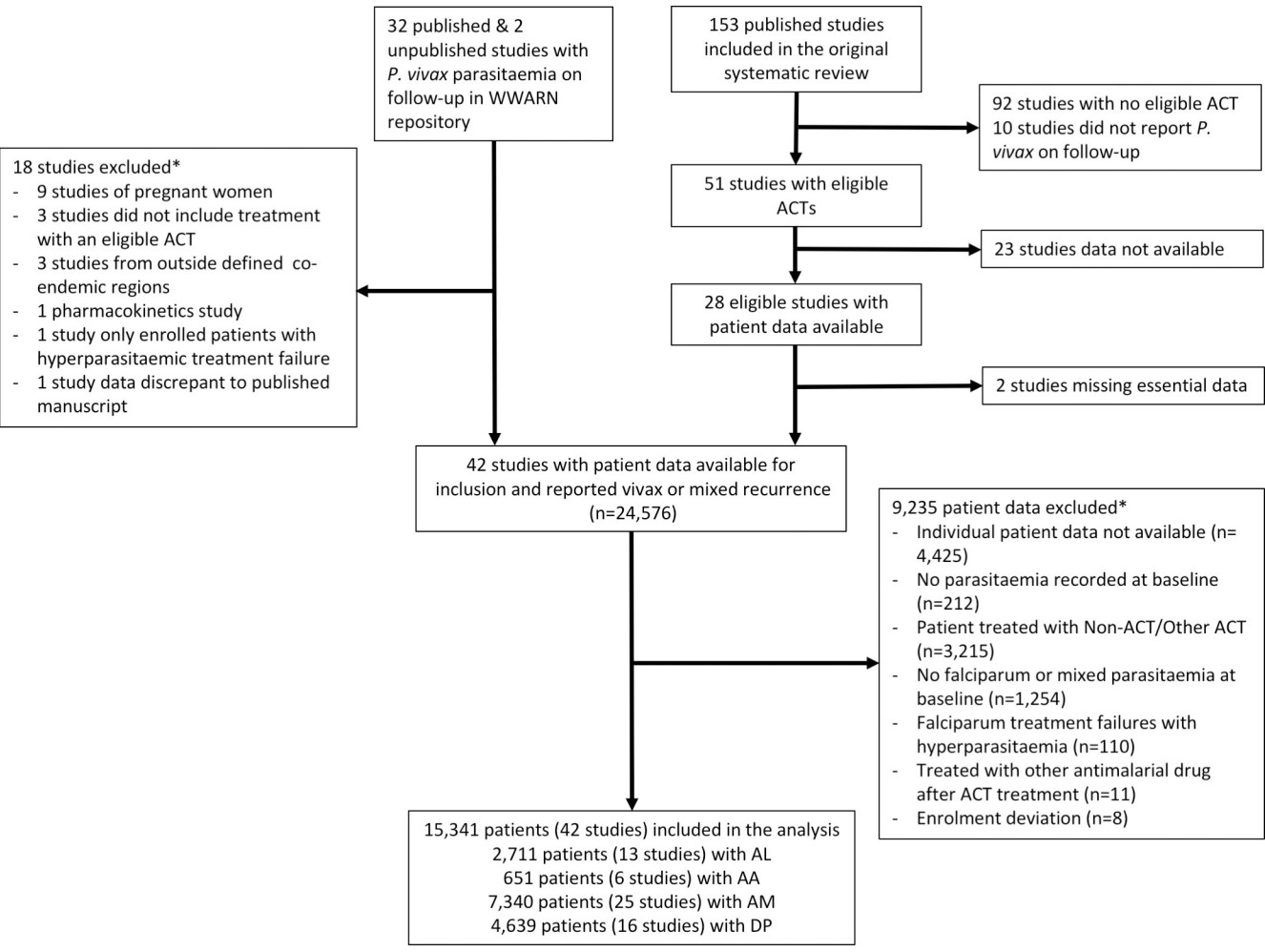

**Fig 1. Study flowchart.** AA, artesunate-amodiaquine; ACT, artemisinin-based combination therapy; AL, artemether-lumefantrine; AM, artesunate-mefloquine; DP, dihydroartemisinin-piperaquine; WWARN, WorldWide Antimalarial Resistance Network.

treated with 1 of the 4 main ACTs (Table 1). The median parasite density at baseline was 7,544/μL (IQR = 2,000–28,021; range = 7–479,280). Baseline Hb data were available for 12,184 (79.4%) patients, of whom 27.6% (3,358) were anaemic (Hb <10 g/dL). No issues impacting data integrity were identified during collation of individual patient data. Studies that were targeted but for which data were not available were more likely to have been conducted in Africa; these studies had a similar median age and sex distribution to those included (S4 Table and S5 Table). Study quality was generally high (S1 Text).

## The overall risk of *P. vivax* parasitaemia

Recurrent parasitaemia between day 7 and 42 was recorded in 2,020 (13.2%) patients, of whom 1,047 (51.8%) had *P. vivax* monoinfection, 99 (4.9%) mixed *P. vivax* and *P. falciparum* infection, and 874 (43.3%) *P. falciparum* monoinfection. The overall cumulative risks of recurrent parasitaemia of any species were 8.1% (95% CI 7.7–8.6) by day 28, 16.8% (16.2–17.5) by day 42, and 30.5% (29.4–31.6) by day 63. The corresponding cumulative risks of *P. vivax* parasitaemia (monoinfection or mixed infection) were 3.7% (3.4–4.1), 10.2% (9.6–10.7), and 22.2%

**Table 1. Demographics and baseline characteristics.**

| | AL, n (%) (n = 2,711) | AA, n (%) (n = 651) | AM, n (%) (n = 7,340) | DP, n (%) (n = 4,639) | Overall, n (%) (n = 15,341) |
|---|---|---|---|---|---|
| Sex* | | | | | |
| Female | 958 (35.4%) | 252 (38.8%) | 3,015 (41.1%) | 1,391 (30.2%) | 5,616 (36.7%) |
| Male | 1,752 (64.6%) | 398 (61.2%) | 4,323 (58.9%) | 3,214 (69.8%) | 9,687 (63.3%) |
| Age, years† | | | | | |
| Median (IQR) | 18.0 (9.0–28.0) | 16.0 (8.0–28.0) | 14.0 (8.0–25.0) | 22.0 (12.0–32.0) | 17.0 (9.0–29.0) |
| ≥15 | 1,596 (58.9%) | 352 (54.2%) | 3,535 (48.2%) | 3,275 (70.8%) | 8,758 (57.1%) |
| 5 to <15 | 753 (27.8%) | 252 (38.8%) | 2,994 (40.8%) | 984 (21.3%) | 4,983 (32.5%) |
| <5 | 361 (13.3%) | 45 (6.9%) | 810 (11.0%) | 368 (8.0%) | 1,584 (10.3%) |
| Weight, kg‡ | | | | | |
| Median (IQR) | 44.0 (21.0–52.0) | 38.0 (20.5–49.4) | 35.9 (18.3–49.0) | 42.5 (19.0–53.0) | 39.0 (19.0–50.0) |
| 5 to <15 | 350 (13.3%) | 67 (10.3%) | 1,008 (14.6%) | 443 (16.6%) | 1,868 (14.5%) |
| 15 to <25 | 418 (15.8%) | 134 (20.7%) | 1,575 (22.8%) | 444 (16.7%) | 2,571 (20.0%) |
| 25 to <35 | 237 (9.0%) | 80 (12.3%) | 783 (11.3%) | 210 (7.9%) | 1,310 (10.2%) |
| 35 to <45 | 361 (13.7%) | 126 (19.4%) | 1,004 (14.5%) | 309 (11.6%) | 1,800 (14.0%) |
| 45 to <55 | 833 (31.6%) | 155 (23.9%) | 1,693 (24.5%) | 677 (25.4%) | 3,358 (26.1%) |
| 55 to <80 | 438 (16.6%) | 86 (13.3%) | 837 (12.1%) | 569 (21.4%) | 1,930 (15.0%) |
| ≥80 | 2 (0.1%) | 0 (0.0%) | 9 (0.1%) | 12 (0.5%) | 23 (0.2%) |
| Relapse periodicity | | | | | |
| Long | 120 (4.4%) | 202 (31.0%) | 499 (6.8%) | 262 (5.6%) | 1,083 (7.1%) |
| Short | 2,591 (95.6%) | 449 (69.0%) | 6,841 (93.2%) | 4,377 (94.4%) | 14,258 (92.9%) |
| Geographical region | | | | | |
| Asia-Pacific | 2,591 (95.6%) | 651 (100.0%) | 6,918 (94.3%) | 4,377 (94.4%) | 14,537 (94.8%) |
| The Americas | 0 (0.0%) | 0 (0.0%) | 422 (5.7%) | 262 (5.6%) | 684 (4.5%) |
| Africa | 120 (4.4%) | 0 (0.0%) | 0 (0.0%) | 0 (0.0%) | 120 (0.8%) |
| Treatment supervision | | | | | |
| Not specified | 67 (2.5%) | 202 (31.0%) | 397 (5.4%) | 0 (0.0%) | 666 (4.3%) |
| Supervised | 1,843 (68.0%) | 294 (45.2%) | 5,987 (81.6%) | 2,182 (47.0%) | 10,306 (67.2%) |
| Partial supervised | 801 (29.5%) | 155 (23.8%) | 956 (13.0%) | 2,457 (53.0%) | 4,369 (28.5%) |
| *P. falciparum* gametocytes present§ | 238/1,959 (12.1%) | 93/428 (21.7%) | 697/5,765 (12.1%) | 495/3,568 (13.9%) | 1,523/11,720 (13.0%) |
| Mixed infection | 183/2,711 (6.8%) | 42/651 (6.5%) | 712/7,340 (9.7%) | 258/4,639 (5.6%) | 1,195/15,341 (7.8%) |
| Baseline parasitaemia¶ | | | | | |
| Median (IQR) | 9,120 (1,900–39,600) | 8,680 (2,364–24,507) | 7,213 (1,976–25,622) | 7,574 (2,048–25,120) | 7,544 (2,000–28,022) |
| ≤100,000 parasites per µL | 2,507 (92.5%) | 638 (98.0%) | 6,610 (94.2%) | 4,290 (94.5%) | 14,045 (94.2%) |
| >100,000 parasites per µL | 204 (7.5%) | 13 (2.0%) | 404 (5.8%) | 250 (5.5%) | 871 (5.8%) |
| Hb, g/dL; mean (SD)‖ | 10.4 (2.7) | 10.7 (2.4) | 10.8 (2.5) | 10.9 (2.7) | 10.8 (2.6) |
| Anaemic, Hb < 10 g/dL | 721/2,586 (27.9%) | 212/530 (40.0%) | 1,655/6,481 (25.5%) | 770/2,587 (29.8%) | 3,358/12,184 (27.6%) |
| Fever, temperature >37.5˚C** | 1,331/2,547 (52.3%) | 332/440 (75.5%) | 3,713/6,565 (56.6%) | 1,099/2,221 (49.5%) | 6,475/11,773 (55.0%) |

*Data not available for 38 patients.

†Data not available for 16 patients.

‡Data not available for 2,481 patients.

§Data not available for 3,623 patients.

¶Parasite count data not available for 427 patients.

‖Data not available for 3,159 patients.

**Data not available for 3,570 patients.

**Abbreviations:** AA, artesunate-amodiaquine; AL, artemether-lumefantrine; AM, artesunate-mefloquine; DP, dihydroartemisinin-piperaquine; Hb, haemoglobin; IQR, interquartile range; n, number; SD, standard deviation.

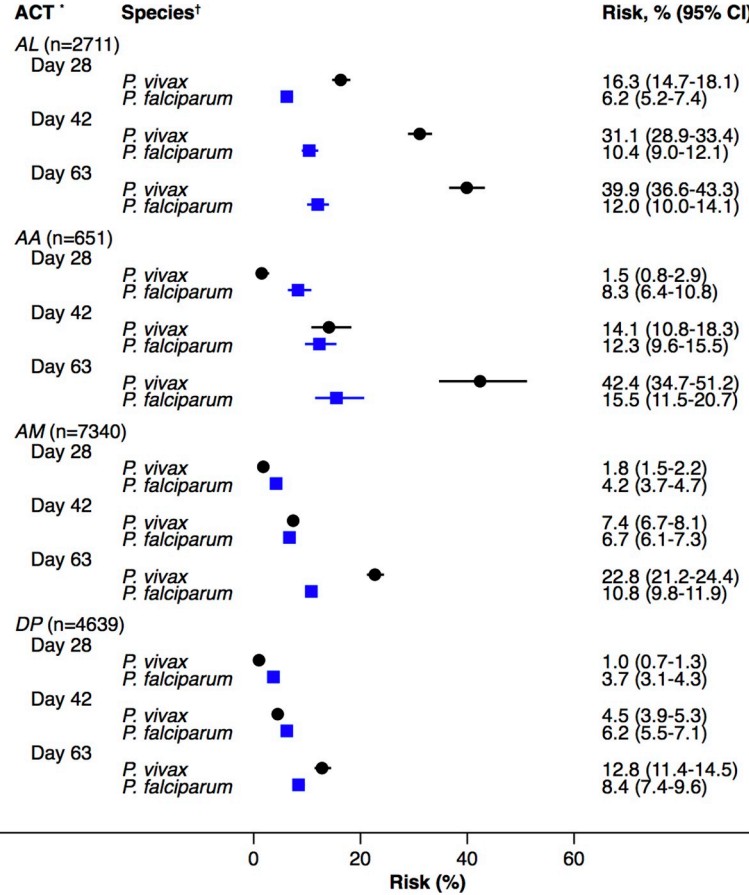

**Fig 2. Cumulative risk (Kaplan–Meier analysis) of *P. vivax* parasitaemia following ACTs.** *ACTs (AA, AL, AM, DP). †*P. vivax* recurrence includes recurrences with *P. vivax* monoinfection or mixed-species infection. AA, artesunate-amodiaquine; ACT, artemisinin-based combination therapy; AL, artemether-lumefantrine; AM, artesunate-mefloquine; DP, dihydroartemisinin-piperaquine.

(21.1–23.3), respectively, and for *P. falciparum* parasitaemia (monoinfection) were 4.5% (4.2–4.9), 7.3% (6.9–7.8), and 10.4% (9.8–11.2), respectively.

At day 42, the cumulative risk of *P. vivax* parasitaemia was 31.1% (95% CI 28.9–33.4; prediction interval) after AL, 14.1% (10.8–18.3) after AA, 7.4% (6.7–8.1) after AM, and 4.5% (3.9–5.3) after DP. By day 63, the risk of *P. vivax* parasitaemia had risen to 39.9% (36.6–43.3) after AL, 42.4% (34.7–51.2) after AA, 22.8% (21.2–24.4) after AM, and 12.8% (11.4–14.5) after DP (Fig 2). There was substantial heterogeneity in risk between studies (Fig 3, Fig 4, Fig 5, Fig 6 and S6 Table). Of the 39 studies in which patients were followed for at least 42 days, the risk of *P. vivax* parasitaemia was ≥20% in 75.0% (9/12) of studies following AL, 20.0% (1/5) following AA, 21.7% (5/23) following AM, and 25.0% (4/16) following DP.

## Risk factors for *P. vivax* parasitaemia

The rate of *P. vivax* parasitaemia was greatest in children. Compared with adults, children under 5 years had a hazard ratio (HR) = 4.40, 95% CI 3.66–5.29; $p < 0.001$, and children 5 to <15 years had an HR = 2.19, 1.90–2.53; $p < 0.001$. Other significant univariable baseline risk factors for *P. vivax* parasitaemia were high parasite count (>100,000 parasites per μL) (HR = 1.54, 1.24–1.90; $p < 0.001$), mixed *P. falciparum* and *P. vivax* infection (HR = 2.54,

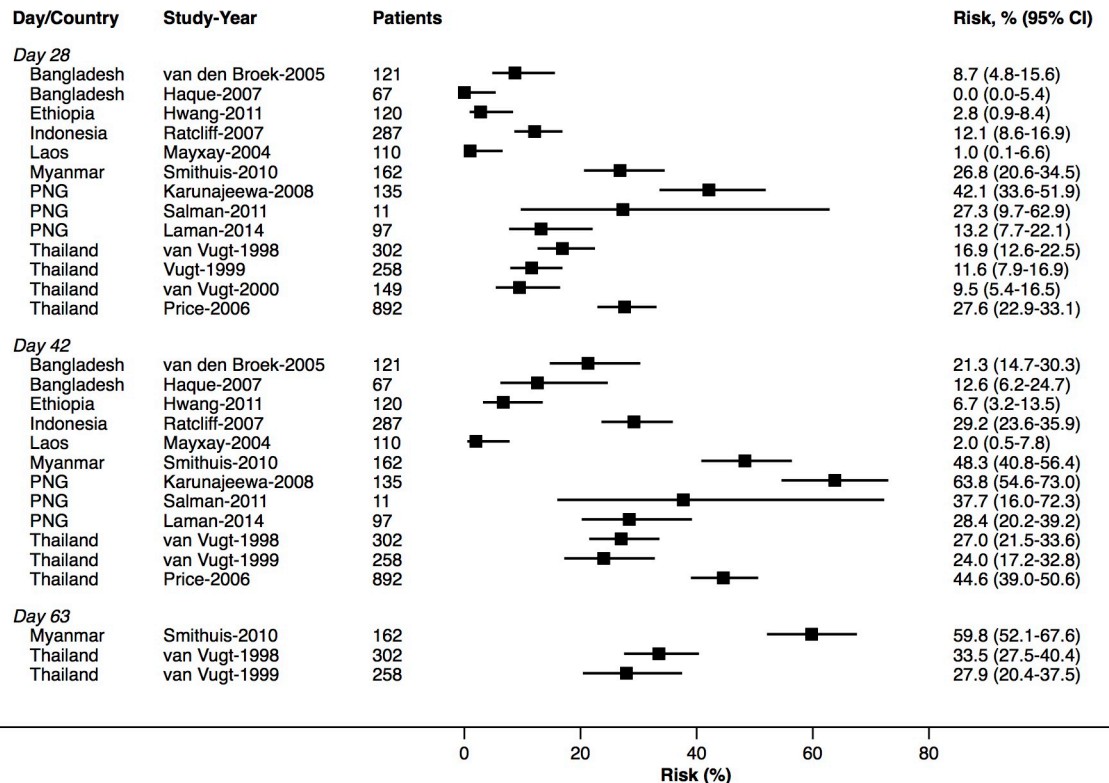

**Fig 3. Cumulative risk (Kaplan–Meier analysis) of *P. vivax* parasitaemia after *P. falciparum* infection by study for AL. AL, artemether-lumefantrine.**

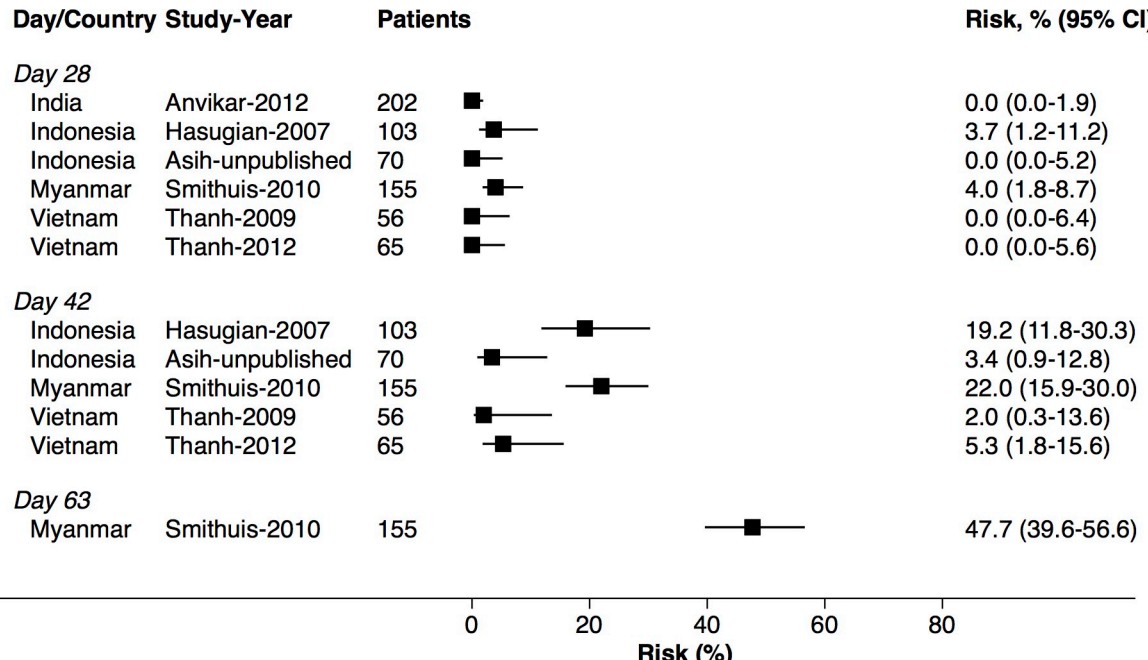

**Fig 4. Cumulative risk (Kaplan–Meier analysis) of *P. vivax* parasitaemia after *P. falciparum* infection by study for AA. AA, artesunate-amodiaquine.**

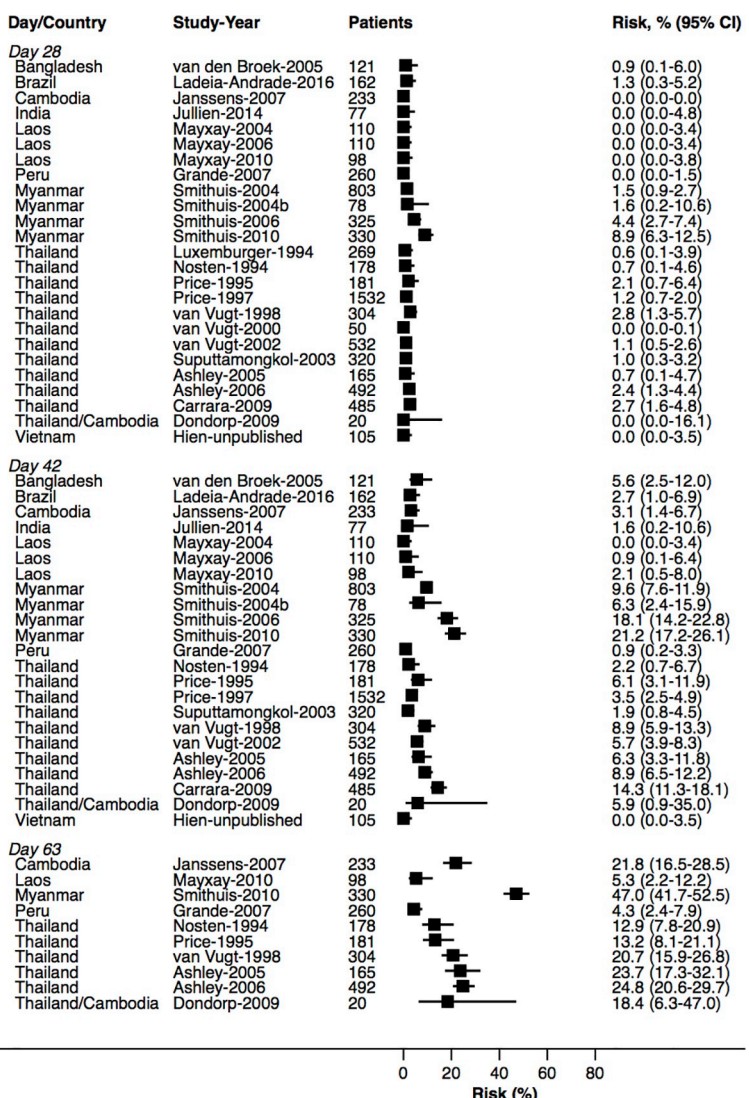

**Fig 5. Cumulative risk (Kaplan–Meier analysis) of *P. vivax* parasitaemia after *P. falciparum* infection by study for AM. AM, artesunate-mefloquine.**

2.14–3.01; p < 0.001), and *P. falciparum* gametocytaemia (HR = 1.57, 1.24–1.99; p < 0.001). Increasing baseline Hb was a protective factor (HR = 0.88 per 1 g/dL increase, 0.86–0.90; p < 0.001). Patients enrolled into studies conducted in regions of short relapse periodicity had a significantly greater rate of *P. vivax* parasitaemia than those enrolled into studies conducted in regions of long relapse periodicity (HR = 8.61, 2.34–31.65; p = 0.001) (Table 2).

In multivariable analysis, younger afge (adjusted HR [AHR] = 3.04, 95% CI 2.39–3.87, p < 0.001 and AHR = 1.81, 95% CI 1.52–2.15, p < 0.001 comparing age <5 years and age 5 to <15 years to adults, respectively), short relapse periodicity (AHR = 6.20, 95% CI 1.98–19.47; p = 0.002), *P. falciparum* gametocytaemia (AHR = 1.40, 1.10–1.79; p = 0.007), mixed infection (AHR = 2.20, 1.79–2.70; p < 0.001), high parasite count (AHR = 1.59, 1.22–2.08; p = 0.001), male sex (AHR = 1.26, 95% CI 1.08–1.46; p = 0.003), and low baseline Hb (AHR = 0.94 per 1 g/dL increase, 0.90–0.97; p < 0.001) were independent risk factors for *P. vivax* parasitaemia (Table 2). Compared with patients treated with DP, the rate of *P. vivax* parasitaemia was

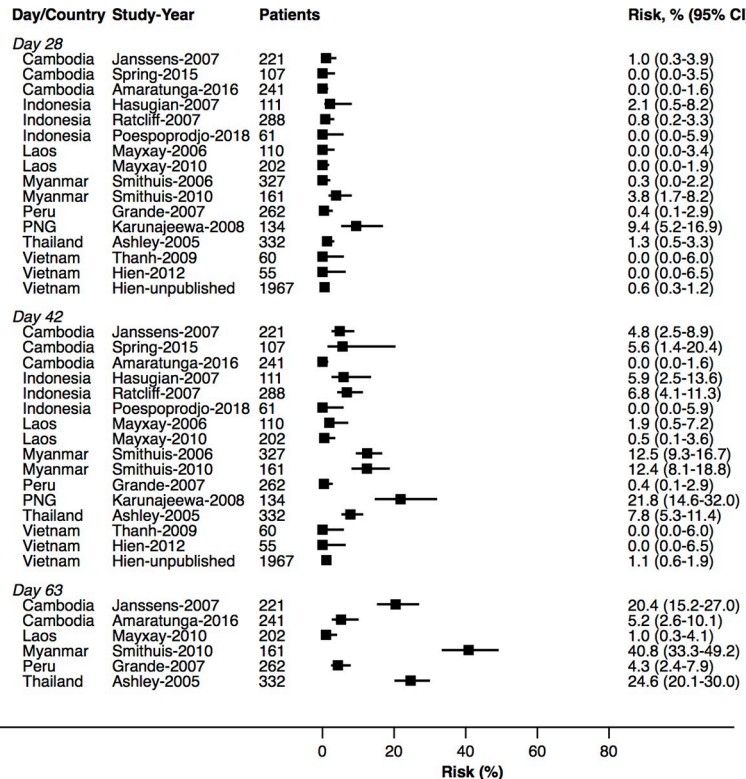

**Fig 6. Cumulative risk (Kaplan–Meier analysis) of *P. vivax* parasitaemia after *P. falciparum* infection by study for DP. DP, dihydroartemisinin-piperaquine.**

greater after AL (AHR = 6.23, 4.55–8.53; p < 0.001), AA (AHR = 2.26, 1.40–3.65; p = 0.001), and AM (AHR = 1.40, 1.04–1.89; p = 0.028) (Fig 7). Removal of 1 study at a time did not reveal bias related to any individual study (S7 Table).

The mean Hb concentration at baseline was 11.8 g/dL (standard deviation [SD] 2.35) compared with 10.9 g/dL (SD 2.03) on day 7 in 6,328 patients with available data. The overall mean fall in Hb between baseline and day 7 was 0.87 g/dL (SD 1.71, range −8.8 to 11.6). The rate of *P. vivax* parasitaemia did not correlate with the absolute fall in Hb in univariable or multivariable analysis.

## Association between risk of *P. vivax* parasitaemia and parasite clearance of initial *P. falciparum* infection

Overall, 46.3% (4,260/9,208) of patients were aparasitaemic by day 1 and 89.9% (8,277/9,208) by day 2; S8 Table. The risk of *P. vivax* parasitaemia between days 7 and 63 was 16.3% (95% CI: 14.7–18.1) in the patients clearing their parasitaemia on day 1, 22.6% (95% CI: 20.7–24.5) in patients clearing on day 2, and 29.2% (95% CI: 25.1–33.7) in patients clearing after this. The rate of *P. vivax* parasitaemia by day 63 was investigated in a multivariable model controlling for confounding factors. Compared with patients clearing their peripheral *P. falciparum* parasites by day 1, the rate of *P. vivax* was significantly greater in those clearing their initial parasitaemia on day 2 (AHR = 1.48, 95% CI 1.27–1.73; p < 0.001) and even greater in those clearing their parasitaemia thereafter (AHR = 1.82, 1.44–2.30; p < 0.001) (Fig 8 and S9 Table). Removal of 1 study at a time did not reveal bias related to any individual study (S10 Table).

**Table 2. Risk factors for *P. vivax* parasitaemia after falciparum infection between days 7 and 42 in patients with different ACTs.**

| | Total Patients | Patients with *P. vivax* Parasitaemia | Risk of *P. vivax* Parasitaemia at Day 42 | Univariable Analyses | | Multivariable Analyses* | |
|---|---|---|---|---|---|---|---|
| | | | | Crude HR (95% CI) | p-Value | Adjusted HR (95% CI) | p-Value |
| Sex | | | | | | | |
| Male | 9,687 | 714 | 10.32 (9.62–11.06) | 1.10 (0.97–1.24) | 0.125 | 1.26 (1.08–1.46) | 0.003 |
| Female | 5,616 | 426 | 9.94 (9.08–10.88) | Reference | | Reference | |
| Age, per year increase | 15,325 | 1,140 | | 0.95 (0.95–0.96) | <0.001 | – | |
| Age category, years | | | | | | | |
| <5 | 1,584 | 313 | 24.95 (22.63–27.45) | 4.40 (3.66–5.29) | <0.001 | 3.04 (2.39–3.87) | <0.001 |
| 5 to <15 | 4,983 | 462 | 11.90 (10.92–12.96) | 2.19 (1.90–2.53) | <0.001 | 1.81 (1.52–2.15) | <0.001 |
| ≥15 | 8,758 | 365 | 6.01 (5.44–6.64) | Reference | | Reference | |
| Weight per kg increase | 12,860 | 1,116 | | 0.97 (0.96–0.97) | <0.001 | – | |
| Relapse periodicity | | | | | | | |
| Short | 14,258 | 1,125 | 10.85 (10.27–11.47) | 8.61 (2.34–31.65) | 0.001 | 6.20 (1.98–19.47) | 0.002 |
| Long | 1,083 | 15 | 1.73 (1.05–2.87) | Reference | | Reference | |
| Geographical region | | | | | | | |
| Africa | 120 | 7 | 6.65 (3.23–13.46) | 0.27 (0.03–2.65) | 0.261 | – | |
| The Americas | 684 | 7 | 1.11 (0.53–2.31) | 0.08 (0.01–0.74) | 0.026 | – | |
| Asia-Pacific | 14,537 | 1,126 | 10.74 (10.16–11.35) | Reference | | – | |
| *P. falciparum* gametocytes | | | | | | | |
| Yes | 1,523 | 222 | 17.84 (15.81–20.09) | 1.57 (1.24–1.99) | <0.001 | 1.40 (1.10–1.79) | 0.007 |
| No | 10,197 | 575 | 8.12 (7.50–8.79) | Reference | | Reference | |
| Mixed infection | | | | | | | |
| Yes | 1,195 | 184 | 21.82 (19.15–24.80) | 2.54 (2.14–3.01) | <0.001 | 2.20 (1.79–2.70) | <0.001 |
| No | 14,146 | 956 | 9.21 (8.67–9.79) | Reference | | Reference | |
| Parasitaemia, parasites per μL | | | | | | | |
| Every 10-times increase | 14,916 | 1,134 | | 1.31 (1.20–1.41) | <0.001 | – | |
| >100,000 parasites/μL | 871 | 112 | 17.68 (14.90–20.91) | 1.54 (1.24–1.90) | <0.001 | 1.59 (1.22–2.08) | 0.001 |
| ≤100,000 parasites/μL | 14,045 | 1,022 | 9.93 (9.36–10.52) | Reference | | Reference | |
| Hb every 1 g/dL increase | 12,184 | 1,090 | | 0.88 (0.86–0.90) | <0.001 | 0.94 (0.90–0.97) | <0.001 |
| Anaemic, Hb < 10 g/dL | | | | | | | |
| Yes | 3,358 | 482 | 18.55 (17.09–20.10) | 1.70 (1.49–1.94) | <0.001 | – | |
| No | 8,826 | 608 | 8.94 (8.29–9.65) | Reference | | – | |
| Drug | | | | | | | |
| AL | 2,711 | 522 | 31.08 (28.89–33.40) | 5.18 (4.10–6.56) | <0.001 | 6.23 (4.55–8.53) | <0.001 |
| AA | 651 | 47 | 14.07 (10.77–18.27) | 1.75 (1.19–2.55) | 0.004 | 2.26 (1.40–3.65) | 0.001 |
| AM | 7,340 | 423 | 7.36 (6.71–8.06) | 1.37 (1.09–1.72) | 0.006 | 1.40 (1.04–1.89) | 0.028 |
| DP | 4,639 | 148 | 4.50 (3.85–5.27) | Reference | | Reference | |

*Geographical region and weight excluded from multivariable analysis due to collinearity with relapse periodicity and age.

**Abbreviations:** AA, artesunate-amodiaquine; ACT, artemisinin-based combination therapy; AL, artemether-lumefantrine; AM, artesunate-mefloquine; CI, confidence interval; DP, dihydroartemisinin-piperaquine; Hb, haemoglobin; HR, hazard ratio.

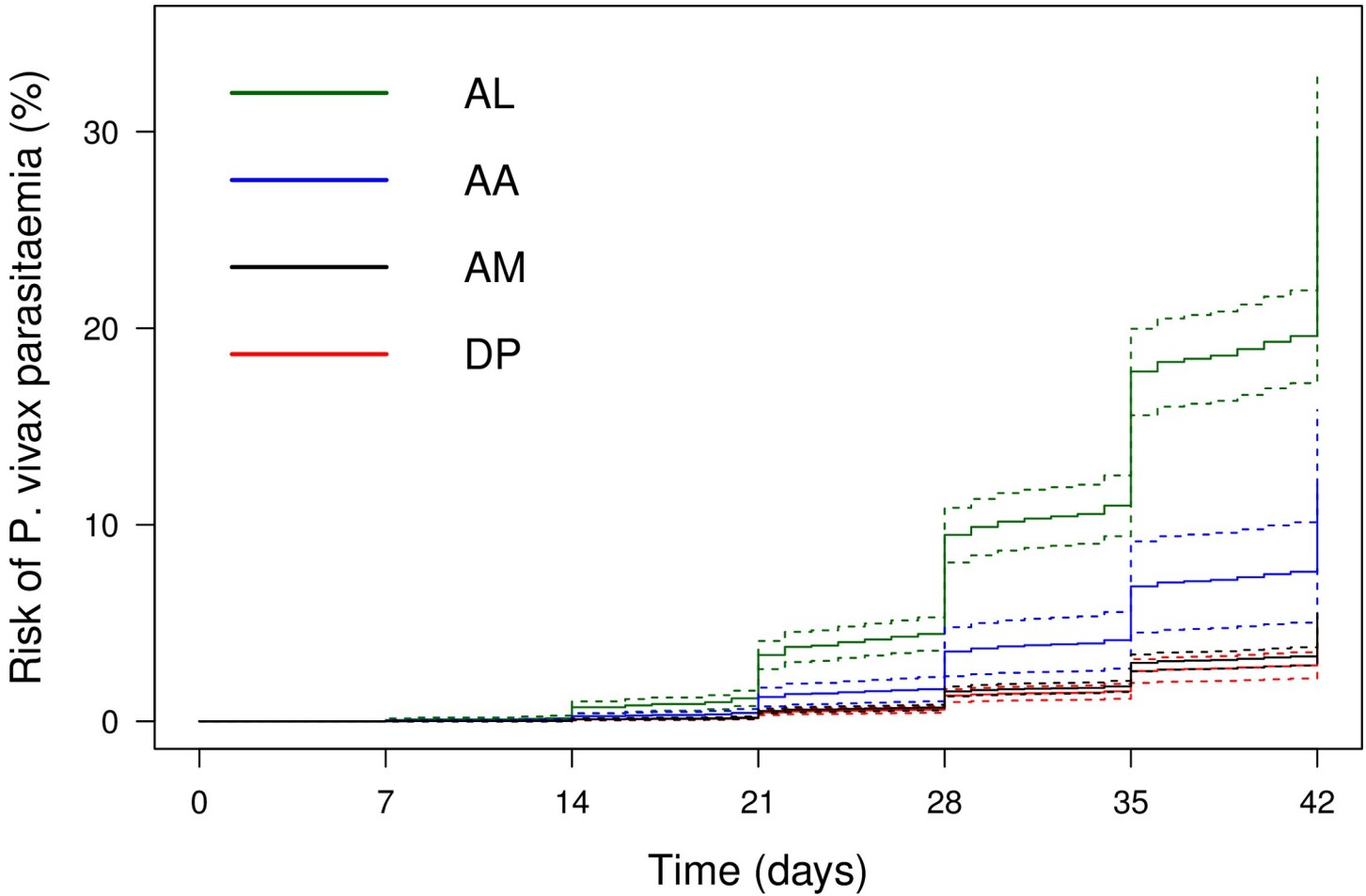

**Fig 7. Risk of *P. vivax* parasitaemia following falciparum infection between day 7 and day 42 according to treatment.** Dotted lines demonstrate 95% CIs.
Figure adjusted for age, sex, baseline parasitaemia, regional relapse periodicity, *P. falciparum* gametocytes, mixed infection, and baseline Hb, assuming no study-site effect.
AA, artesunate-amodiaquine; AL, artemether-lumefantrine; AM, artesunate-mefloquine; DP, dihydroartemisinin-piperaquine; Hb, haemoglobin.

A sensitivity analysis including patients treated with single low-dose primaquine did not change these results substantially (compared with patients who cleared parasitaemia by day 1, AHR = 1.48 [95% CI 1.27–1.73] if cleared on day 2 and AHR = 1.82 [95% CI 1.44–2.29] if cleared thereafter).

## Site factors associated with risk of *P. vivax* parasitaemia following treatment

The study-specific risks of *P. vivax* during follow-up are presented in Fig 3, Fig 4, Fig 5 and Fig 6. The risk of *P. vivax* parasitaemia at day 42 was greatest following treatment with AL. Twelve studies across 18 study sites and 7 countries enrolled and treated 2,562 patients with AL who were followed for 42 days or longer (S2 Fig). Nine (75%) studies enrolled children younger than 5 years. The study-specific risk of *P. vivax* at day 42 varied from 0% to 63.8%. The risk was ≥20% in study sites from Thailand (4 study sites), Papua New Guinea (PNG) (4 study sites), Myanmar (3 study sites), and Indonesia (1 study site); 12%–26% in 3 study sites from Bangladesh; 0%–7% in 2 study sites from Ethiopia; and 2.0% in 1 study from Laos (Fig 3). The corresponding data for the other treatment arms are presented in Fig 4, Fig 5, Fig 6 and S11 Table.

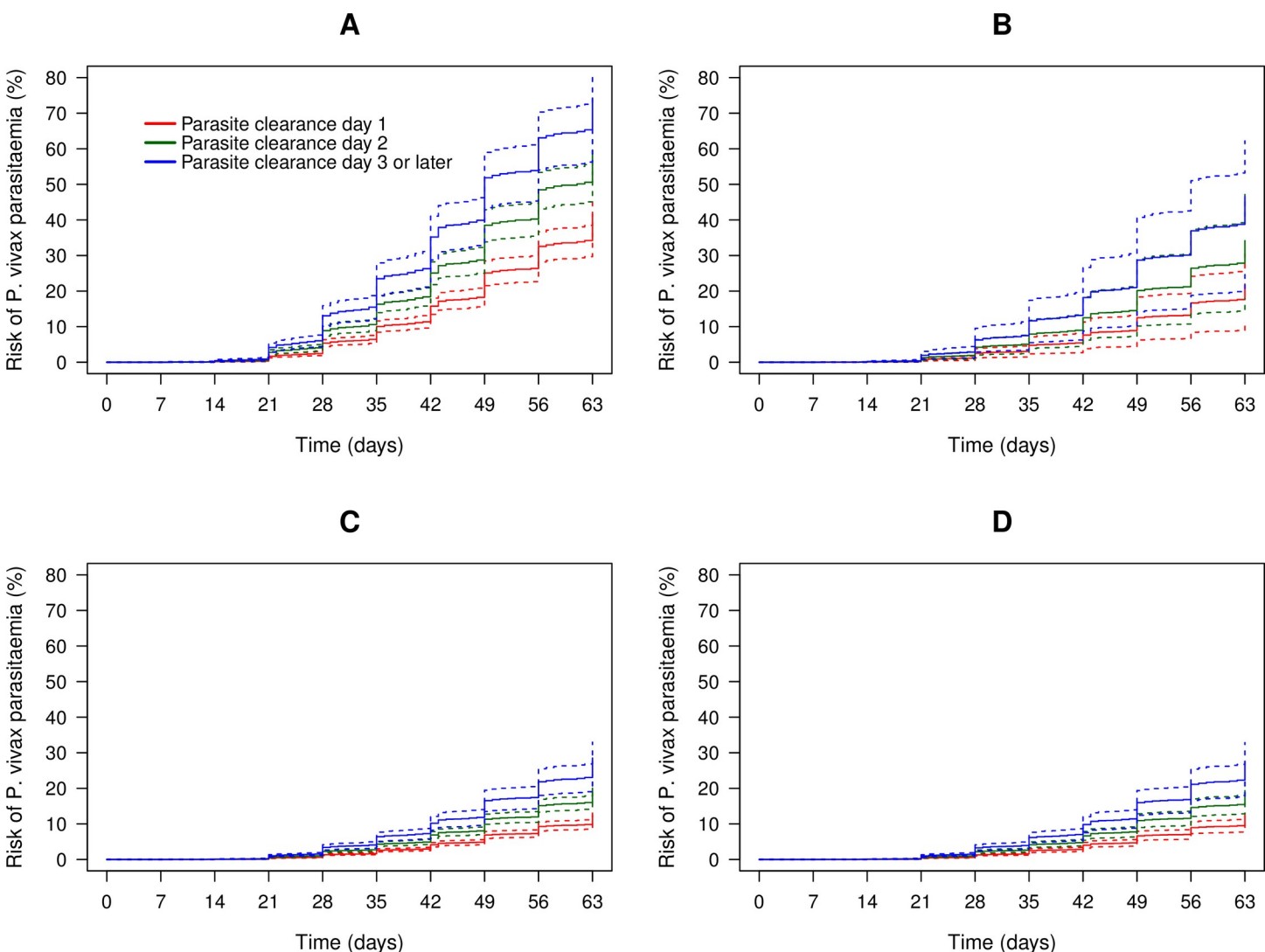

**Fig 8. Risk of *P. vivax* parasitaemia following falciparum infection between days 7 and 63 according to treatment and day of parasite clearance: (A) AL, (B) AA, (C) AM, and (D) DP.** Figure adjusted for age, sex, baseline parasitaemia, regional relapse periodicity, *P. falciparum* gametocytes, mixed infection at baseline, and Hb at baseline, assuming no study-site effect. AA, artesunate-amodiaquine; AL, artemether-lumefantrine; AM, artesunate-mefloquine; DP, dihydroartemisinin-piperaquine; Hb, haemoglobin.

Using subnational data, the estimated incidence of *P. falciparum* at each site varied from 0.5 to 154 cases per 1,000 person-years, and the background incidence of *P. vivax* varied from 1.6 to 151 cases per 1,000 person-years. The risk of *P. vivax* following AL was correlated with the site-specific background incidence of *P. vivax* ($r_s$ = 0.676, p = 0.0029) and incidence of *P. falciparum* ($r_s$ = 0.607, p = 0.0098) (Fig 9). However, after controlling for confounding factors in a Cox regression model, neither of the background incidences were associated significantly with the risk of *P. vivax* (AHR = 0.99, 0.97–1.02, p = 0.634 and AHR = 1.00, 0.99–1.02, p = 0.629, respectively, for every 1 case increase per 1,000 person-years) (S12 Table). The results were similar for the other ACTs (S13 Table, S14 Table, S15 Table, S3 Fig, S4 Fig, S5 Fig). The year of enrolment was not independently associated with the risk of *P. vivax* (AHR = 0.93 per year later, 0.84–1.03; p = 0.173).

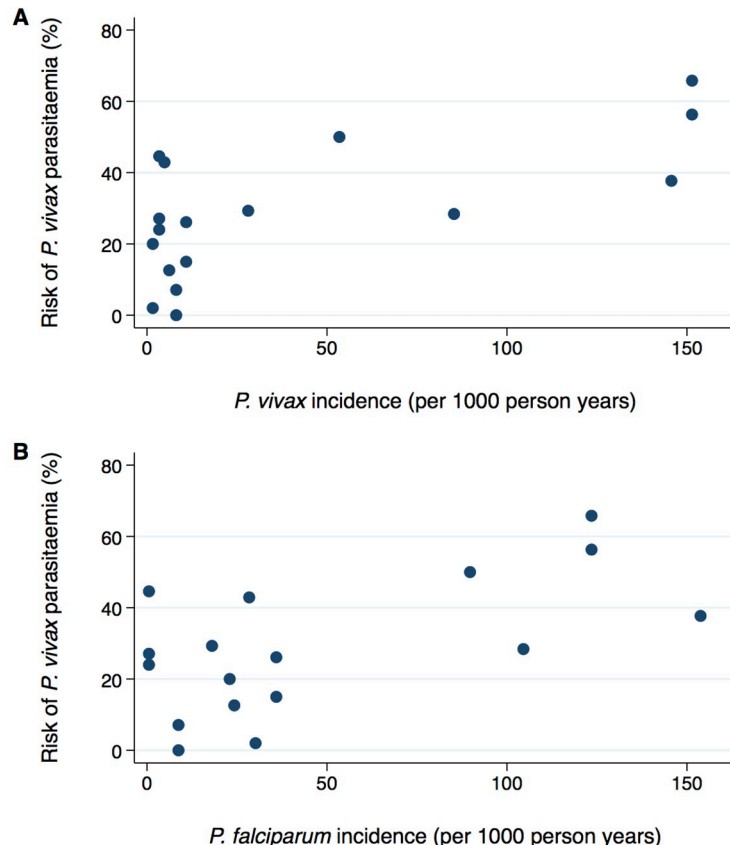

**Fig 9. Risk of *P. vivax* parasitaemia at day 42 following treatment with AL according to background subnational incidence of *P. vivax* (A) and *P. falciparum* (B).** $r_s$ = 0.676, p = 0.0029 (A) and $r_s$ = 0.607, p = 0.0098 (B). AL, artemether-lumefantrine.

## Discussion

Our individual patient data meta-analysis, including more than 15,000 patients, confirms a high risk of *P. vivax* parasitaemia following treatment of *P. falciparum* with ACTs; however, there was marked heterogeneity between study populations and sites. The risk of *P. vivax* was highest in studies undertaken in regions with short relapse periodicity and in patients who were young or presented with mixed-species infections or a high baseline parasitaemia.

Overall, 10% of patients treated with an ACT had *P. vivax* recurrence by day 42, and this increased to 22% by day 63. The risk of *P. vivax* was significantly lower in patients treated with an ACT containing a slowly eliminated partner drug such as piperaquine, mefloquine, and amodiaquine [5, 7]. Lumefantrine is eliminated faster, with a half-life of 4 days, and after 16 days (4 half-lives) provides minimal post-treatment prophylaxis against *P. vivax* relapses [59]. This is reflected by the high risk of *P. vivax* parasitaemia following AL, which reached 31% by day 42.

The heterogeneity in the risk of *P. vivax* was highlighted in our recent systematic review of clinical trials undertaken in coendemic areas [7] and has been investigated further in a review of trials from the eastern border of Myanmar [60]. On the Thailand–Myanmar border, the risk of *P. vivax* parasitaemia within 42 days of treatment of falciparum malaria exceeded 20% between 2003 and 2010 and reached 85% in 1 study. However, after 2010, the risk fell to less

than 5%. The authors hypothesise that the decline in the risk of *P. vivax* was attributable to substantial improvements in malaria control and thus a decline in the proportion of patients presenting with *P. falciparum* who were coinfected with *P. vivax*. A key determinant of *P. vivax* relapse is the number of hypnozoite parasites present in the liver, which varies with local *P. vivax* transmission intensity, immunity, and sporozoite inoculum [61]. Although the risk of *P. vivax* was correlated positively with the predicted background incidence of both *P. vivax* and *P. falciparum*, this was not apparent after adjusting for confounding factors. We observed no temporal trends indicative of a lower risk of *P. vivax* parasitaemia in more recent studies. This may relate to the relatively few studies available to compare the use of an individual ACT at a single site, lack of temporally distinct data from multiple studies in the same regions, or heterogeneity between study populations from different locations. Incidence estimates are also inexact, relying on subnational regional data that may not equate to the study site and do not account for seasonal variation of transmission [15]. There is also a risk of inclusion bias, with prospective clinical efficacy studies generally undertaken in regions with a relatively high malaria burden to ensure adequate recruitment. The ability to compare the effect of malaria prevalence between clinical efficacy studies is therefore likely to be limited compared with data from longitudinal cohorts [62].

Our individual patient data meta-analysis confirms an increased risk of recurrence in regions where the duration between *P. vivax* relapses is short [5, 7] and highlights the additional risks of young age, high baseline parasitaemia, low baseline Hb, and presenting with either mixed-species infection or *P. falciparum* gametocytes. More than 15% of patients presenting with mixed infection of *P. falciparum* and *P. vivax* had *P. vivax* parasitaemia within 42 days, supporting the World Health Organization (WHO) recommendation for the use of primaquine radical cure in these patients. However, the high risk of *P. vivax* was also apparent in those presenting with *P. falciparum* monoinfection. We could not determine whether recurrences were due to relapse or reinfection in patients continuing to reside in a malaria-endemic area. The subnational incidence estimates of *P. vivax* infection in the study sites included in our study ranged from 0.2 to 151 per 1,000 person-years [15]. If one conservatively assumes 2 weeks post-treatment prophylaxis following antimalarial treatment and the highest risk of reinfection of the included study sites, then the maximum risk of a new infection with *P. vivax* within 63 days of *P. falciparum* would be 2.0%. The risks of *P. vivax* parasitaemia in our analysis were thus at least 6- to 21-fold–greater than could be explained by reinfection alone.

It is possible that patients with mixed infections of *P. falciparum* with low-level *P. vivax* parasitaemia were misdiagnosed as *P. falciparum* monoinfections. However, all of the ACTs included in the analysis have high efficacy against *P. vivax* [63], and even if 10% of patients had had submicroscopic *P. vivax* parasitaemia at presentation, the risk of recrudescent *P. vivax* by day 63 would be no greater than 0.5%. Conversely, we may have underestimated the risk of vivax recurrence because submicroscopic relapses were not quantified and the duration of follow-up was restricted to 63 days, thus preventing detection of later relapses. In a longitudinal cohort in which asymptomatic submicroscopic infections were quantified over a 12-month period, there was a 2.4-fold–higher risk of asymptomatic vivax recurrence following asymptomatic falciparum infections [64].

Our findings are more consistent with the *P. vivax* parasitaemia during follow-up arising from activation of *P. vivax* hypnozoites present at the time of the initial presentation with *P. falciparum* [5, 65, 66]. Importantly, the initial therapeutic response, as measured by parasite clearance, was correlated with the subsequent risk of *P. vivax*. Patients taking longer to clear their *P. falciparum* parasitaemia were at almost 2-fold–greater risk of *P. vivax* than those who cleared their parasitaemia rapidly. Whilst this could reflect vulnerability of a host with low immunity to both slower parasite clearance and recurrent parasitaemia, the increased risk of *P.*

*vivax* was independent of the patients' age, a surrogate marker of host immunity. We believe this supports the hypothesis that vivax parasitaemia following falciparum malaria relates to a host–parasite interaction that activates hypnozoites acquired from a prior infection. Haemolysis has been postulated to trigger *P. vivax* reactivation [66, 67], although we found no correlation between the early fall in Hb concentration and the risk of subsequent vivax parasitaemia.

Our study has a number of limitations. The systematic review included all *P. falciparum* clinical trials between 1960 and 2018, although the analysis was restricted to patients who were treated with an ACT with studies conducted between 1991 and 2018. The analysis included 42 studies, enrolling 15,341 patients in 12 malaria-endemic countries, and the quality of the included studies suggested a low risk of systematic bias (S1 Text); however, there were significant differences in these data and those included in our previous systematic review [7]. Fourteen studies uploaded to the WWARN repository with *P. vivax* recurrences were not included as part of the 153 studies in the previous review. Ten of these were not included in the initial systematic review because the corresponding manuscripts reported neither the presence nor absence of vivax recurrences. An additional limitation of our current analysis was the inability to include data from 25 targeted studies. Although the patient demographics from these studies were similar to the included studies, they included a greater number of African studies, which might have allowed better generalisability of results to this region. Furthermore, the studies that were included were not equally representative of all vivax-endemic locations, with the majority undertaken in southeast Asia and only a single study included from India. Studies in the WWARN repository were only included into the analysis if they reported a *P. vivax* recurrence in at least 1 treatment arm. This was justified because in many *P. falciparum* studies, *P. vivax* parasitaemia may not be deemed relevant to the efficacy analysis, and data may not be recorded or shared with WWARN. Hence, it is possible that exclusion of these may have systematically biased against studies without any *P. vivax* recurrence and thus overestimated the risk of *P. vivax*. Reassuringly, only 3 out of 33 clinical trial arms in an independent systematic review reported no *P. vivax* recurrences [60].

The rapid elimination of *P. vivax* will require a greater emphasis on addressing the hidden reservoirs of the parasite, including individuals with asymptomatic carriage of blood- or liver-stage parasites. Our analysis highlights that in coendemic areas, patients presenting with *P. falciparum* may be at significant risk of carrying dormant hypnozoites. Hence, whilst current WHO guidelines only recommend treatment with a prolonged course of primaquine for patients with *P. vivax* or a *P. vivax*–mixed-species infection [68], consideration should be given to offering radical cure, with either primaquine or tafenoquine, with appropriate G6PD testing, to patients presenting with *P. falciparum* monoinfection in coendemic regions. In view of the heterogeneity in the risk of *P. vivax* and drug-induced haemolysis between locations, such a strategy should be recommended with caution. The risk factors for *P. vivax* identified in this analysis may help define populations for whom the benefit of a universal policy of radical cure has greatest benefit but should be guided by complementary prospective clinical efficacy studies, and in coendemic areas, these should include quantifying the subsequent risk of all species of *Plasmodia* following antimalarial treatment.

## Supporting information

**S1 PRISMA Checklist. PRISMA-IPD.** PRISMA-IPD, Preferred Reporting Items for Systematic Review and Meta-Analyses of individual participant data
(PDF)

**S1 Box. Search strategy.**
(PDF)

**S1 Text. Assessment of risk of bias relating to individual studies.**
(PDF)

**S1 Fig. Map of study-site locations.** Map created using ggplot2 in R.
(TIF)

**S2 Fig. Map of cumulative risk of *P. vivax* parasitaemia at day 42 following AL.** Map created using ggplot2 in R. AL, artemether-lumefantrine
(TIF)

**S3 Fig. Risk of *P. vivax* parasitaemia at day 42 following treatment with AA according to subnational (A) *P. vivax* incidence and (B) *P. falciparum* incidence.** AA, artesunate-amodiaquine
(PDF)

**S4 Fig. Risk of *P. vivax* parasitaemia at day 42 following treatment with AM according to subnational (A) *P. vivax* incidence and (B) *P. falciparum* incidence.** AM, artesunate-mefloquine
(PDF)

**S5 Fig. Risk of *P. vivax* parasitaemia at day 42 following treatment with DP according to subnational (A) *P. vivax* incidence and (B) *P. falciparum* incidence.** DP, dihydroartemisinin-piperaquine
(PDF)

**S1 Table. Reasons for studies not being included in analysis.**
(PDF)

**S2 Table. Studies included in analysis.**
(PDF)

**S3 Table. Study sites included in analysis.**
(PDF)

**S4 Table. Studies targeted for the analysis but not available.**
(PDF)

**S5 Table. Comparison of baseline characteristics from included and targeted studies.**
(PDF)

**S6 Table. Prediction intervals for cumulative risk of *P. vivax* parasitaemia following ACTs.** ACT, artemisinin-based combination therapy
(PDF)

**S7 Table. Sensitivity analysis for associations between patient characteristics and rate of *P. vivax* parasitaemia between days 7 to 42 for the general model.**
(PDF)

**S8 Table. Parasite clearance according to treatment.**
(PDF)

**S9 Table. Relationship between day of parasite clearance, patient characteristics, and rate of *P. vivax* parasitaemia between days 7 and 63.**
(PDF)

**S10 Table. Sensitivity analysis for associations between patient characteristics and rate of *P. vivax* parasitaemia between days 7 to 63 for the model including day of parasite clearance.**
(PDF)

**S11 Table. Studies with follow-up for 42 days or longer included in analysis of site factors.**
(PDF)

**S12 Table. Relationship between patient characteristics and study-site malaria prevalence and rate of *P. vivax* parasitaemia between days 7 and 42 in patients treated with AL.** AL, artemether-lumefantrine
(PDF)

**S13 Table. Relationship between patient characteristics and study-site malaria prevalence and rate of *P. vivax* parasitaemia between days 7 and 42 in patients treated with AA. AA, artesunate-amodiaquine.**
(PDF)

**S14 Table. Relationship between patient characteristics and study-site malaria prevalence and rate of *P. vivax* parasitaemia between days 7 and 42 in patients treated with AM. AM, artesunate-mefloquine.**
(PDF)

**S15 Table. Relationship between patient characteristics and study-site malaria prevalence and rate of *P. vivax* parasitaemia between days 7 and 42 in patients treated with DP. DP, dihydroartemisinin-piperaquine.**
(PDF)

## Acknowledgments

We thank all patients and staff who participated in these clinical trials at all the sites and the WWARN team for technical and administrative support. We also thank Katherine Battle for providing the estimates of subnational incidence of *P. vivax* and *P. falciparum* clinical malaria.

The opinions expressed are those of the authors and do not necessarily reflect those of the Australian Defence Force, Joint Health Command, or any extant Australian Defence Force policy. The findings and conclusions in this report are those of the author(s) and do not necessarily represent the official position of the US Centers for Disease Control and Prevention. The views expressed here are solely those of the authors and do not reflect the views, policies, or positions of the US Government or Department of Defense. Material has been reviewed by the Walter Reed Army Institute of Research. There is no objection to its presentation and/or publication. The opinions or assertions contained herein are the private views of the author and are not to be construed as official or as reflecting true views of the Department of the Army or the Department of Defense. The investigators have adhered to the policies for protection of human subjects as prescribed in Army Regulation 70–25.

## Author Contributions

**Conceptualization:** Mohammad S. Hossain, Robert J. Commons, Nicholas M. Douglas, Kamala Thriemer, Kasia Stepniewska, Philippe J. Guerin, Julie A. Simpson, Ric N. Price.

**Data curation:** Mohammad S. Hossain, Robert J. Commons, Kasia Stepniewska.

**Formal analysis:** Mohammad S. Hossain, Robert J. Commons, Nicholas M. Douglas, Julie A. Simpson, Ric N. Price.

**Funding acquisition:** Ric N. Price.

**Investigation:** Kamala Thriemer, Bereket H. Alemayehu, Chanaki Amaratunga, Anupkumar R. Anvikar, Elizabeth A. Ashley, Puji B. S. Asih, Verena I. Carrara, Chanthap Lon, Umberto D'Alessandro, Timothy M. E. Davis, Arjen M. Dondorp, Michael D. Edstein, Rick M. Fairhurst, Marcelo U. Ferreira, Jimee Hwang, Bart Janssens, Harin Karunajeewa, Jean R. Kiechel, Simone Ladeia-Andrade, Moses Laman, Mayfong Mayxay, Rose McGready, Brioni R. Moore, Ivo Mueller, Paul N. Newton, Nguyen T. Thuy-Nhien, Harald Noedl, Francois Nosten, Aung P. Phyo, Jeanne R. Poespoprodjo, David L. Saunders, Frank Smithuis, Michele D. Spring, Seila Suon, Yupin Suputtamongkol, Din Syafruddin, Hien T. Tran, Neena Valecha, Michel Van Herp, Michele Van Vugt, Nicholas J. White, Ric N. Price.

**Methodology:** Mohammad S. Hossain, Robert J. Commons, Kasia Stepniewska, Julie A. Simpson, Ric N. Price.

**Project administration:** Kasia Stepniewska, Philippe J. Guerin, Ric N. Price.

**Resources:** Kasia Stepniewska, Philippe J. Guerin, Julie A. Simpson, Ric N. Price.

**Writing – original draft:** Mohammad S. Hossain, Robert J. Commons, Nicholas M. Douglas, Kamala Thriemer, Julie A. Simpson, Ric N. Price.

**Writing – review & editing:** Mohammad S. Hossain, Robert J. Commons, Nicholas M. Douglas, Kamala Thriemer, Bereket H. Alemayehu, Chanaki Amaratunga, Anupkumar R. Anvikar, Elizabeth A. Ashley, Puji B. S. Asih, Verena I. Carrara, Chanthap Lon, Umberto D'Alessandro, Timothy M. E. Davis, Arjen M. Dondorp, Michael D. Edstein, Rick M. Fairhurst, Marcelo U. Ferreira, Jimee Hwang, Bart Janssens, Harin Karunajeewa, Jean R. Kiechel, Simone Ladeia-Andrade, Moses Laman, Mayfong Mayxay, Rose McGready, Brioni R. Moore, Ivo Mueller, Paul N. Newton, Nguyen T. Thuy-Nhien, Harald Noedl, Francois Nosten, Aung P. Phyo, Jeanne R. Poespoprodjo, David L. Saunders, Frank Smithuis, Michele D. Spring, Kasia Stepniewska, Seila Suon, Yupin Suputtamongkol, Din Syafruddin, Hien T. Tran, Neena Valecha, Michel Van Herp, Michele Van Vugt, Nicholas J. White, Philippe J. Guerin, Julie A. Simpson, Ric N. Price.

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
