## [Editor Report · Decision Letter 0]

1 Jun 2020

Dear Dr Price, 

Thank you for submitting your manuscript entitled "The risk of Plasmodium vivax parasitaemia after Plasmodium falciparum malaria: a WWARN individual patient data meta-analysis" for consideration by PLOS Medicine.

Your manuscript has now been evaluated by the PLOS Medicine editorial staff and I am writing to let you know that we would like to send your submission out for external peer review.

Kind regards,

Artur Arikainen,

Associate Editor

PLOS Medicine

---

## [Decision Letter · Decision Letter 1]

25 Jun 2020

Dear Dr. Price,

Thank you very much for submitting your manuscript "The risk of Plasmodium vivax parasitaemia after Plasmodium falciparum malaria: a WWARN individual patient data meta-analysis" (PMEDICINE-D-20-02368R1) for consideration at PLOS Medicine. 

[LINK]

In light of these reviews, I am afraid that we will not be able to accept the manuscript for publication in the journal in its current form, but we would like to consider a revised version that addresses the reviewers' and editors' comments. Obviously we cannot make any decision about publication until we have seen the revised manuscript and your response, and we plan to seek re-review by one or more of the reviewers. 

We expect to receive your revised manuscript by Jul 16 2020 11:59PM. Please email us (plosmedicine@plos.org) if you have any questions or concerns.

We look forward to receiving your revised manuscript. 

Sincerely,

Emma Veitch, PhD

PLOS Medicine

On behalf of Clare Stone, PhD, Acting Chief Editor,

PLOS Medicine

plosmedicine.org

*Reviewer 1 queried whether first-person phrasing was accepted by the journal (eg, "our analysis"; "we did"), just to inform the authors, we absolutely do accept and often will prefer this type of writeup as it is more direct and succinct.

*At this stage, we ask that you include a short, non-technical Author Summary of your research to make findings accessible to a wide audience that includes both scientists and non-scientists. The Author Summary should immediately follow the Abstract in your revised manuscript. This text is subject to editorial change and should be distinct from the scientific abstract. Please see our author guidelines for more information: https://journals.plos.org/plosmedicine/s/revising-your-manuscript#loc-author-summary

Comments from the reviewers:

Reviewer #1: See attachment

Reviewer #2: See attachment

Michael Dewey

Reviewer #3: General comments / Major issues

This is a very well written paper on an important topic in malaria epidemiology that may have implications for Plasmodium vivax (Pv) elimination strategies. The study aims to define the risk factors of Pv parasitemia following a Pf infection in various settings using an individual-level meta-analysis. While this is a well done and interesting meta-analysis, there are number of comments the authors should address. 

Major compulsory comments to be addressed:

* My main issue with the paper, as currently written, is a lack of explanation as to the rationale for the study. It is clear the authors hypothesis that a Pf infection, and subsequent treatment, increases the risk of a relapse from a concurrent hypnozoite Pv infection, but the authors offer little rationale as to why they believe this hypothesis to be true. Is it simply because in areas with both Pf and Pv transmission, individuals with a Pf infection would also be at greater risk for harboring a Pv infection because of myriad factors in low and heterogeneous transmission that put specific people at greater risk? Would it have to do with immunity? Declines in hemoglobin following Pf treatment are explored, but are not stipulated a priori as a potential reason for Pv reactivation. Such rationales for the primary hypothesis under study should be clearly stated in the introduction, tested in the methods and results, and then reassessed and interpreted in the discussion.

* One of the major weakness of this study is disentangling what the level of Pv relapse means among individuals with a Pf infection that are treated with various ACTs (e.g. cumulative risk of Pv parasitemia were 3.7%, 10.2% and 22.2%, at days 28, 42 and 63 respectively - lines 322-324). But how does this compare to the underlying rate of Pv parasitemia/relapse at these time points among individuals without a Pf infection who never received treatment? I.e. - What is the additional risk of Pv relapse due to a Pf-treated infection compared to those without a Pf infection? I understand the authors do not have such data on individuals without a Pf infection, but that comparison is what is needed to actually test if the rate of relapse in Pv is higher than what would be expected among those without a Pf clearance. This limitation needs to be clearly stated and discussed. Can the authors offer any insight from the literature into how they think Pv parasitemia/activation levels would differ between those with and without a Pf infection and treatment? 

* Could one factor that might explain the potential increased risk of Pv relapse among individuals treated for a Pf infection be the underlying level of Pf vs Pf transmission risk in the study sites - perhaps measured as the ration of Pf:Pv infections at baseline? If areas with much more Pv transmission relative to Pf transmission had higher risk of Pv relapse as compared to areas with higher Pf to Pv transmission at specified follow-ups, to me this would indicate that those with a Pf infection are just at higher risk of any Plasmodium infection. Can the authors include a Pf:Pv ratio indicator, perhaps categorized, as a covariate in the Cox PH models assessing the risk factors of Pv parasitemia? Assuming this could be derived by MAP estimates of PfPR and PvPR for each study site?

* Can the authors more clearly state in the discussion the major limitation of the study that they cannot distinguish between a new Pv infection and a reactivation of a previous infection? Are the levels of parasitemia after 28, 42 and 63 days in line with transmission risk for a new Pv infection over this period in the study sites? I doubt they are and this argument should be made that they are likely reactivations because they are much higher than transmission would likely allow for them to be new Pv infections over these relatively short time periods?

* Did the Cox models also account for malaria transmission season? They should if they have not.

* I would think the underlying level of risk of transmission of Pf and Pv would be a significant risk factor (or potential confounder) when assessing the risk of a Pf infection treated with various ACTs. The authors appear to try and account for this with a covariate in the models for malaria parasite prevalence from MAP (lines 230-231). Why was this then categorized as low / not low at a cut-point of PfPR = 1.5%? As defined, this was not significant in the Cox models? Did they try other cut-points or categories? Why not just keep PfPR continuous - to me that would be the best option for accounting for this very important factor in the analyses. 

* The authors state a date range of 1960 - 2018 was used for the systematic review of Pf treatment studies, taken for a previous study. This is a bit disingenuous as ACTS did not become available until well after 1960 (I would imagine in the 1990s at the earliest)? The authors should revise the date ranges for the systematic review accordingly, as well as provide the date ranges of the earliest and latest included studies in their description of the systematic review results. 

* It would be helpful if the authors could state how the covariates in the Cox PH models were defined (e.g. those in lines 257 - 259). For example, was mixed infection categorized as yes/no (0/1). Was hemoglobin a continuous variable?

* For the analysis assessing the site-specific risk factors associated with risk of Pv following treatment (section starting on line 378), why was this limited to those treated with AL? Why not include all ACTs and just account for ACT type with a fixed effect covariate in the model?

[LINK]

---

## [Decision Letter · Decision Letter 2]

24 Aug 2020

Dear Dr. Price,

Thank you very much for re-submitting your manuscript "The risk of Plasmodium vivax parasitaemia after Plasmodium falciparum malaria: a WWARN individual patient data meta-analysis" (PMEDICINE-D-20-02368R2) for review by PLOS Medicine.

I have discussed the paper with my colleagues and the academic editor and it was also seen again by three reviewers. I am pleased to say that provided the remaining editorial and production issues are dealt with we are planning to accept the paper for publication in the journal.

[LINK]

We look forward to receiving the revised manuscript by Aug 31 2020 11:59PM. 

Sincerely,

Artur Arikainen

Associate Editor

PLOS Medicine

plosmedicine.org

Requests from Editors:

1. Title: Please update to: “The risk of Plasmodium vivax parasitaemia after Plasmodium falciparum malaria: an individual patient data meta-analysis from the WorldWide Antimalarial Resistance Network”

2. Short Title: Please update to: “Risk of P. vivax after P. falciparum malaria”

3. Funding Statement: Please replace the last sentence with our standard text, if applicable: “The funders had no role in study design, data collection and analysis, decision to publish, or preparation of the manuscript.”

4. Competing Interests Statement: All authors must declare their relevant competing interests per the PLOS policy, which can be seen here:

https://journals.plos.org/plosmedicine/s/competing-interests For authors with ties to industry, please indicate whether any of the interests has a financial stake in the results of the current study. Please therefore include employment by author(s) to AstraZeneca. Please also add this statement to the manuscript's Competing Interests: "EAA and NJW are Academic Editors on PLOS Medicine's editorial board."

5. Abstract:

a. Please report your abstract according to PRISMA for abstracts: http://www.plosmedicine.org/article/info:doi/10.1371/journal.pmed.1001419 . Please include the databases searched, the method used to assess quality/risk of bias, a summary of the quality/risk of bias assessment, and the key funding source(s).

b. Please summarise the numbers of each study type included, eg. RCTs, cohort studies.

c. Please summarise the number of participants by drug combination (from around line 340), and by region (from Table 1).

d. Please add the following summary data for the patients from the main Results section: “Overall 14,146 (92.2%) patients had P. falciparum mono-infection and 1,195 (7.8%) mixed infection with P. falciparum and P. vivax confirmed by blood film microscopy. The median age was 17.0 years (inter-quartile-range (IQR) = 9.0 – 29.0 years; range = 0 – 80 years), with 1,584 (10.3%) patients younger than 5 years.”

e. Please quantify all results with 95% CIs and p values.

f. Line 138: Please start with "In this study, we found that ..." or similar

g. Line 140 (and 169): Please replace “effective” with “recommended” or similar, to avoid implying causality from your findings.

6. Author Summary:

a. Line 149: Please clarify “parasitaemia” for a lay reader.

b. Lines 152, 168, 169: Please clarify “hypnozoite” for a lay reader, or replace with a simpler term.

c. Lines 161-162: Please clarify “short relapse periodicity” for a lay reader.

d. Line 164: Please give species names here as elsewhere, for clarity.

e. Line 165: Perhaps replace “is … heterogeneity” with “are … differences”, for simplicity.

f. Line 171: Perhaps replace “complementary” with “further”, for simplicity.

7. Please move your citation callout to before punctuation, with a space after the text, eg.: “associated with 405,000 deaths [1].”

8. The terms gender and sex are not interchangeable (as discussed in http://www.who.int/gender/whatisgender/en/ ); please use the appropriate term.

9. Lines 609-648: Please remove this information (Contributions, COIs, Funding, and Data) from the main text and instead include it in the online submission form.

10. Lines 651-655: Please move the ethics information to the Methods.

11. Please provide full access details (eg. DOI or URL) for references 1, 2, 10, and 68. Please remove this from reference 63: “following competing interests: NJW is a member of the Editorial Board of PLOS Medicine.”

12. When completing the PRISMA checklist, please use section and paragraph numbers, rather than page numbers.

---------------

Comments from Reviewers:

Reviewer #1: Author responded to all issues I raised. I would recommend publish this manuscript in PLOS Medicine.

Reviewer #2: The authors have addressed my points.

Michael Dewey

Reviewer #3: The authors did a very nice job in addressing the reviewers comments. The manuscript is now suitable for publication.

[LINK]

---

## [Editor Report · Decision Letter 3]

25 Sep 2020

Dear Prof Price, 

On behalf of my colleagues and the academic editor, Dr. James G. Beeson, I am delighted to inform you that your manuscript entitled "The risk of Plasmodium vivax parasitaemia after Plasmodium falciparum malaria: an individual patient data meta-analysis from the WorldWide Antimalarial Resistance Network" (PMEDICINE-D-20-02368R3) has been accepted for publication in PLOS Medicine. 

PRODUCTION PROCESS

PRESS

PROFILE INFORMATION

Thank you again for submitting the manuscript to PLOS Medicine. We look forward to publishing it. 

Best wishes, 

Artur Arikainen, 

Senior Editor 

PLOS Medicine

plosmedicine.org